# Optical Coherence Tomography Identifies Visual Pathway Involvement Earlier than Visual Function Tests in Children with MRI-Verified Optic Pathway Gliomas

**DOI:** 10.3390/cancers14020318

**Published:** 2022-01-09

**Authors:** Urszula Arnljots, Maria Nilsson, Ulrika Sandvik, Ida Hed Myrberg, Daniel Martin Munoz, Klas Blomgren, Kerstin Hellgren

**Affiliations:** 1St. Erik Eye Hospital, 171 64 Stockholm, Sweden; 2Department of Clinical Neuroscience, Karolinska Institutet, 171 76 Stockholm, Sweden; maria.nilsson@ki.se (M.N.); ulrika.sandvik@ki.se (U.S.); kerstin.hellgren@ki.se (K.H.); 3Department of Neurosurgery, Karolinska University Hospital Solna, 171 64 Stockholm, Sweden; 4Unit of Biostatistics, Institute of Environmental Medicine, Karolinska Institutet, 171 77 Stockhom, Sweden; ida.hed.myrberg@ki.se; 5Department of Neuroradiology and Pediatric Radiology, Karolinska University Hospital, 171 64 Stockholm, Sweden; daniel.martin-munoz@regionstockholm.se; 6Department of Women’s and Children’s Health, Karolinska Institutet, 171 77 Stockholm, Sweden; klas.blomgren@ki.se; 7Department of Pediatric Oncology, Karolinska University Hospital, 171 64 Stockholm, Sweden; 8Astrid Lindgren Children’s Hospital, 171 64 Stockholm, Sweden

**Keywords:** optic pathway glioma, optical coherence tomography, ganglion cell-inner plexiform layer (GC-IPL), children, MRI, visual field, children

## Abstract

**Simple Summary:**

Retrograde degeneration of the eye’s retinal ganglion cells, causing visual loss and even blindness, is a feared consequence of optic pathway gliomas. Optical coherence tomography (OCT) is a patient-friendly, high-resolution imaging technique enabling objective measurements of the integrity of the retinal ganglion cell layer. Children with optic pathway glioma unable to complete formal visual field testing and/or reliable visual acuity testing, may undergo OCT, providing objective information about visual loss and potential clinical progression. By combining visual functional measurements with OCT findings, the clinical examination will be safer and more reliable. By improving the clinical follow-up of the tumor, the treatment choices can be optimized thereby preventing further visual loss and, in the worst case, blindness.

**Abstract:**

This study investigates whether optical coherence tomography (OCT) could add useful information in the examination of children with optic pathway glioma (OPG) at high risk of developing vision loss. For this purpose, the relationship between ganglion cell-inner plexiform layer (GC-IPL) thickness and visual function, evaluated with tests of visual acuity (VA) and visual field (VF), as well as tumor site according to magnetic resonance imaging (MRI), were examined in a geographically defined group of children with OPG. Methods: Children aged <18 years with OPG underwent ophthalmic examination including VA, VF (Zeiss HFA perimetry) and OCT imaging (Zeiss Cirrus HD-OCT). Results: Out of 51 patients included, 45 provided 77 eyes with MRI-verified OPG, and 19 patients provided 25 eyes without OPG. Significant correlations were found between GC-IPL, VF and VA (*p* < 0.001). The GC-IPL pattern loss corresponded in 95% to VF defects and in 92% to MRI findings. Conclusions: Our study indicates that GC-IPL measures could serve as an early marker of vision-threatening changes related to OPG and as a valuable link between MRI and visual function tests. Thinning of GC-IPL and differences in topography between eyes are strong indicators of and predictive of vision loss related to OPG.

## 1. Introduction

Optic pathway gliomas (OPGs) account for 3–5% of all brain tumors in children (1.15 to 5.14 cases per 100,000, with the highest rates reported in the United States [1]) and affect around 20% of subjects with neurofibromatosis type 1 (NF1) [1,2,3]. They are most often benign pilocytic astrocytomas (WHO grade I in 94.9% [4,5,6,7]. Most children with OPGs typically present between one and eight years of age (65% <5 years of age [8]), a time when reliability and accuracy of vision examination may be compromised by young age, lack of cooperation, as well as misinterpretation of the child’s behavior [9]. Fundoscopic examination may exhibit edema or pallor of the optic disc, however, this part of the examination can be difficult to perform and, in addition, normal fundus findings, do not exclude existing OPGs. Moreover, optic atrophy or disc edema can also be found even if visual acuity (VA) is normal [10]. In such cases, visual field (VF) testing would provide additional information about optic nerve function, but young children are frequently unable to complete this examination [11]. 

Optical coherence tomography (OCT) is a simple, non-invasive imaging technique of the retinal layers. The OCT device admits segmentation in images that gives information about the retina’s different layers [12]. It can be performed within a few minutes and only requires stable fixation for a few seconds, compared to VF examinations which require four to 10 min of fixation. It was introduced into ophthalmological healthcare in 1991 [12]. Since then, it is used in screening, diagnosing, and monitoring diseases of macula and optic nerve. After brain injuries affecting the primary visual pathways, signs of neuro-retinal damage in the eyes can be noticed. OCT has proven to be useful in the diagnosis and follow-up of adult patients with tumors affecting the primary visual pathways [10]. In the same manner, early signs of neural tissue destruction could potentially be quantified with OCT also in children [13]. 

The ganglion cell layer (GCL) is composed of the retinal ganglion cells which receive information from photoreceptors via interneurons. The GCL is thickest and most dense where it forms a circular rim around the macular fovea. Most of the ganglion cell bodies are in this rim. The peripapillary retinal nerve fiber layer (pRNFL) contains the unmyelinated axons of the retinal ganglion cells that merge to form the optic nerve. The macular GCL and the pRNFL thicknesses are assessed by OCT and decreased values represent retinal ganglion cell loss, which occurs through Wallerian degeneration if the lesion is in the anterior visual pathway and through retrograde transsynaptic degeneration in posterior visual pathway lesions [14,15]. 

The retinotopic mapping in the cortex assessed with magnetic resonance imaging (MRI) tractography has been shown to correlate well with the pRNFL and the GCL, assessed with OCT in patients who have suffered pre- or perinatal brain lesions [16]. The pattern of the ganglion cell loss has in a multiple case study been found to correlate well with visual function and the location and extent of brain injury [17]. In addition, data obtained on nine patients with ischemic cortical injury presenting with laterality of VF defect suggest that there was a significant difference in GCL reduction between the affected and unaffected sides but no statistically significant difference between affected and unaffected pRNFL. The measurement of GC-IPL thickness of the macula, rather than the pRNFL, may therefore be a more accurate and reliable marker of vision, demonstrating a stronger relationship with VA and VF [18]. In this context, in children unable to complete perimetry and/or reliable VA testing, OCT can be used as a targeted tool to detect visual dysfunction and objectively measure clinical progression in terms of ganglion cell loss. 

This study aims to investigate whether OCT could add useful information in the examination of children with OPG, particularly those at risk of developing vision loss. For this purpose, the relationship between GC-IPL thickness and visual function, defined as VA and VF, as well as tumor site according to MRI was examined in a geographically defined group of children with OPG. 

## 2. Materials and Methods

All children (≤18 years old) in the Stockholm area with MRI verified OPGs followed at the Astrid Lindgren Children’s Hospital and/or recruited from the Swedish Cancer Register were invited to participate, between January 2017 and January 2020, in a prospective longitudinal study. The ophthalmological examination intervals were planned according to age and clinical progression. In each case both eyes of the patients were examined, if possible. 

### 2.1. Glioma Diagnosis and Location

The diagnosis of OPG was determined using contrast-enhanced MRI. In cases where surgery was performed, biopsy and histopathological analysis were undertaken simultaneously. Tumor location was evaluated by a neurosurgeon (U.S.) and pediatric neuro-radiologist (D.M.), and the area of tumor spread along the visual pathways was shown in a modified model of Dodshgun et al., referred to as PLAN score [19,20]. The date of the analyzed MRI scans was chosen as close to the date of the ophthalmological examination as possible. MRI scans were performed on GE 750w Discovery 3T and GE 450w Optima 1.5T, GE Healthcare, (MIlwaukee, WI, USA), Philips Ingenia 3T and Philips Achieva 1.5T Philips Healthcare system (Best, The Netherlands). Sequences for review included T2 W sagittal, axial and coronal planes, coronal T2 FLAIR and T1 W contrast-enhanced in axial and coronal planes. Depending on the protocol for underlying pathology, slice thickness was 3–4 mm. The intraorbital part of the optic nerve was defined as normal according to normal values in the literature [20,21,22], especially when ectatic and elongated and in the absence of signs of intracranial hypertension, inflammatory or demyelinating disease. Tumor localization and extension according to PLAN score were compared to OCT findings. 

### 2.2. Visual Acuity

Visual acuity and VF were measured for each eye separately. Visual acuity was assessed with Lea symbols (chart design by Hyvärinen [23]) in children <8 years of age and with KM charts (chart design by Konstantin Moutakis [24]) in older children [9]. In children ≥4 years of age visual impairment was classified according to WHO [25] as mild if VA < 0.5, moderate if VA < 0.3, severe if VA < 0.1 VA, and blindness if VA < 0.05 in the better eye. In children <4 years of age, mild visual impairment was not defined but moderate visual impairment was defined as VA < 0.3 and severe visual impairment as VA < 0.1. Refraction was assessed with an autorefractor of eyes under cycloplegia (mixture of phenylephrine 1.5% and cyclopentolate 0.85%). For study purposes refraction was presented as spherical equivalent; SE (combined sphere and 1/2 cylinder). 

### 2.3. Visual Field

Visual field was assessed with HFA (Humphrey^®^ Field Analyzer 3 Carl Zeiss Meditec Inc., Dublin, CA, USA) when possible depending on age and cognitive level [26]. Reliability of each perimetry (false positive errors < 15%, false negative errors < 15% and fixation loss < 25%) was reviewed by two examiners (K.H., U.A.) according to Examiner Based Assessment of Reliability (EBAR) [27]. The VF index (VFI) was recorded. Visual fields were considered pathological when threshold values, representing a probability of <1% of a normal population, showed depression of three or more contiguous points [28]. Up to four isolated abnormal points in the total or pattern deviation plots were acceptable, provided they were not in a cluster and did not involve the same points on retesting [28]. In addition, to simplify the evaluation, VF tests were categorized according to the number of quadrants involved (0–4) and if involvement was complete (in the nasal quadrant ≥10.5/14 and in the temporal quadrant ≥9/12 test points) or partial. See Figure 1 for example of normal VF. 

### 2.4. GC-IPL Layer

Optical coherence tomography images were acquired using a Cirrus HD-OCT device (Cirrus; Carl Zeiss Meditec, Dublin, CA, USA). Macular Cube 512 × 128 or 200 × 200 scan protocol and automated GC-IPL analysis segmentation algorithm incorporated into the Cirrus 6.0 software were selected. The GC-IPL analysis was performed as described in previous studies [29]. All scans were reviewed by three examiners (K.H., M.N., U.A.). Scans with artifacts (28) were not included in the final analysis. Pathological GC-IPL was defined as reduction in average thickness <76 µm (values below the fifth percentile) and/or GC-IPL asymmetry defined as a difference between the thickest and thinnest sector of >10 µm (values above 95th percentile) [29]. Figure 1 shows one example of a normal VF assessed with HFA and neuroretinal structure visualized with OCT as GC-IPL. 

Ophthalmological data were collected from visits at as young age as possible. To best correlate OCT findings with visual function, we selected examinations performed at the same visit. Since reliable HFA was acquired at a higher age than reliable VA and OCT assessments, we used the VA and OCT results from the first visit with a reliable HFA in the analyses that included all three parameters. 

Since pRNFL has been shown to be less accurate than GC-IPL, we only used the latter in the analyses. It is proposed that GC-IPL is a more accurate and reliable biomarker of vision, because it is not confounded by axonal swelling, axonal atrophy, and blood vessel artifacts as in pRNFL analysis [30]. In addition, GC-IPL was found to be far easier to assess and thus yielded more data. The GC-IPL data were compared with VA, VFI and brain MRI findings. Quantitative, statistical analysis as well as quantitative assessments of GC-IPL patterns were used. 

### 2.5. Statistical Analyses

Both eyes of each patient were used for analysis if applicable. Descriptive statistics were presented as means, standard deviations, and frequencies and percentages for categorical variables. Data were visualized using scatterplots. Differences in age at OCT, VFI, and VA, respectively, between children with and without OPG were tested using the Wilcoxon rank sum test. Differences in average GC-IPL, minimum GC-IPL, and VFI between eyes with and without OPG were tested using the Wilcoxon rank sum test for clustered data [31], using function clusWilcox.test in R package clusrank [32]. 

Spearman correlation was used for analyzing correlation between VFI, VA, and average and minimum GC-IPL. All statistical analyses were performed using R version 4.1.1 [33]. 

### 2.6. Quantitative Categorizing Comparisons

In order to evaluate the congruency and degree of overlap between the GC-IPL pattern of thinning on OCT and VF pattern loss a 3-tiered grading approach has been implemented (1: no congruency, 2: partial congruency, 3: complete congruency). Partial congruency was defined as GC-IPL thinning and VF pattern loss that did not show complete congruency or overlap. 

## 3. Results

Fifty-one patients with suspected OPG were found eligible and invited into the study (30 NF1). Five patients (three NF1) had MRI-verified gliomas close to but not affecting the visual pathways. In addition, one patient with NF1 had increased thickness of both optic nerves, not considered to be caused by OPG. In total there were 77 eyes from 45 patients (23 boys, 22 girls; 19 with sporadic and 26 with NF1-associated OPG) and 25 eyes unaffected by OPG from 19 patients, see Figure 2. 

### 3.1. MRI

Mean age at OPG diagnosis was 5.0 years (SD 2.78; range 0.4–11.3) and mean age at study entrance was 9.3 years (SD 4.16; range 2.1–17.6), thus 4.3 years later than diagnosis. In the patients with sporadic OPGs tumor diagnosis was at mean age 4.7 years (SD 2.92; range 0.4–9.0) and in the group with NF1 at mean age 5.2 years (SD 2.71; range 1.3–11.3). Figure 3 presents the locations of all OPGs (longest extension according to lateralization of the tumor). 

### 3.2. Visual Acuity

Visual acuity was evaluated in all 51 patients. There were five children younger than four years of age at the time of the study, three of which had OPG in both eyes. All children were evaluated with symbol charts except for one (aged 2.1 years with OPG) who was evaluated with grating test and the value was converted into decimal values. Regarding VA in the better eye, 40 of the 45 children with OPG had no visual impairment, one child had mild visual impairment, another child had moderate visual impairment, and three children were blind in both eyes (no light perception in two patients). Nine were blind in one eye (no light perception in eight eyes). In total, 15 of the 77 eyes with OPG (19%) were blind. No child without OPG in both eyes was visually impaired in either eye. There was one 3.3-year-old child with unilateral OPG (VA 0.8) with VA0.4 in the contralateral eye. This eye had anisometropic amblyopia. 

There was no significant difference in age at VA testing between patients (*n* = 45; mean 9.3; SD 4.2; range 2.1–17.6 years) with OPG in at least one eye and those without OPG in any eye (*n* = 6; mean 11.3; SD 4.2; range 4.9–17.1 years) *p* = 0.38. There was a significant difference in VA between eyes with OPG (*n* = 77; median 0.7; range 0.0–1.25) and without OPG (*n* = 25; median 1.0, range 0.4–1.25; *p* < 0.001). 

### 3.3. Refraction

Mean SE was −0.3 (SD 1.58, range −10.3 to + 5.0). Mean refraction in eyes with and without OPG was −0.4 (SD 1.79, range −10.3 to +5.0) and −0.1 (SD 0.58; range −2.25 to 1.0), respectively. 

### 3.4. Visual Fields

Thirty-two patients (55 eyes; 54% of all eyes) provided HFA data, of which 40 eyes with OPG and 15 without. Eight children had homonymous VF defects in both eyes. There was no significant difference in age between patients (*n* = 27; mean 11.0; SD 3.5; range 6.3–17.3 years) with OPG in at least one eye and those without OPG (n = 5; mean 13.0; SD 3.4; range 8.3–17.1 years), *p* = 0.360. 

There was a significant difference in VFI between eyes with OPG (*n* = 40; median 88%; range 7–100%) and eyes without OPG (n = 15; median 98%; 89–100%), *p* = 0.001. Twenty-six eyes with OPG and one without OPG had pathological VF according to our definition. 

### 3.5. OCT

Forty-seven patients (84 eyes; 82% of all eyes) provided GC-IPL data, of which 59 eyes were affected by OPG and 25 eyes were without OPG. There was no significant difference in age at OCT examination between patients with OPG in any eye (*n* = 41, mean 9.7; SD 4.1; range 3.3–17.6) and patients without OPG (n = 6, mean 11.3; SD 4.2; range 4.9–17.1) (*p* = 0.50). In the youngest group <4 years, six of 10 eyes did not provide OCT results due to blindness or severe visual impairment (three eyes) and because of young age (three eyes). Median GC-IPL in eyes without OPG was 84.0 (range 72.0–92.0), 70.0 (range 44.0–89.0; *p* < 0.001) in eyes with OPG. The minimum median GC-IPL in eyes without OPG was 83.0 (range 68.0–89.0), 62.0 (range 39–82; *p* < 0.001) in eyes with OPG. Abnormally thin GC-IPL was seen in 48 eyes, of which 46 had OPG and two did not. In 23 of 48 (48%) eyes with average GC-IPL < 76 µm an asymmetry between the thickest and thinnest sector >10 µm was found. No eyes with an asymmetry between the thickest and thinnest sector >10 µm had average GC-IPL ≥ 76 µm. 

### 3.6. OCT, VF, and VA

Figure 4 illustrates the number of examinations performed in all eyes with OPG (A), abnormal results in all eyes with OPG examined (B) and abnormal results among eyes with OPG that underwent all three examinations (C). Forty eyes with OPG yield all three examinations: VA, VF, and GC-IPL measurements. Thin GC-IPL (<76 µm) was seen in 33 of 40 eyes with OPG. There were no eyes that underwent all three examinations and had decline in visual acuity or VF defects without abnormal OCT. The VA was positively correlated with average GC-IPL (*p* < 0.001, Spearman’s ρ 0.6051) and minimum GC-IPL (*p* < 0.001, Spearman’s ρ 0.513). The VFI was positively correlated with average GC-IPL (*p* < 0.001, Spearman’s ρ 0.640) and minimum GC-IPL (*p* < 0.001, Spearman’s ρ 0.725), see Figure 5.

### 3.7. Quantitative Categorizing Comparisons

#### 3.7.1. Optical Coherence Tomography and Visual Field Examinations

In total there were 55 eyes (40 with OPG and 15 without) examined with OCT, VA testing and HFA perimetry. Thirty-three out of 55 eyes (60%) displayed complete congruency between GC-IPL and VF results. Additionally, 19 eyes showed partial congruency (one eye without OPG). In three eyes there were no pattern loss congruencies (GC-IPL changes but no visual field loss with VFI ≥ 90%). 

In 14 out of 15 (93%) eyes without OPG there was complete congruency between OCT pattern and VF pattern. These 14 eyes had no pathology in OCT outcome or visual function. In one eye there was a minor VF defect but normal OCT and VA. 

#### 3.7.2. Optical Coherence Tomography vs. Magnetic Resonance Imaging

We examined the congruence between lateral localization of tumor according to MRI and the pattern of GC-IPL thickness. The tumors were in many cases widespread on MRI, meaning that the involvement of the visual pathways in these cases was partly bilateral and partly unilateral in the same individual. In these cases, we assumed that the tumor affected the visual pathways bilaterally. In cases where the tumors were strictly unilateral on MRI, we judged them to be unilateral, both in the anterior and posterior visual pathways. 

Of the 41 subjects with MRI-verified OPG who had OCT scans, 16 had unilateral tumor spread on MRI (13 patients with isolated optic nerve glioma, two patients with retrochiasmal tumor, and one patient with tumor in the right optic nerve and right chiasm). In 14 of those 16 patients the GC-IPL loss was on the same side as the tumor. In the remaining two patients who both had isolated unilateral optic pathway glioma, the GCL-thickness was normal in both eyes (83 vs. 80 µm) in one and was thin in both eyes (74 vs. 72 µm) in the other. The last-mentioned patient also had anisometropia with myopia. In the 25 subjects with bilateral OPG on MRI, 24 patients had bilateral GC-IPL loss pattern corresponding to the MRI findings. One patient with bilateral OPG (located in left intracanalicular/prechiasmatic segment of left optic nerve with extension to chiasma/optic tract on both sides) had normal GC-IPL thickness in both eyes (80 vs. 80 µm). Thus, in total there were MRI and OCT congruence in 38/41 (92%) of patients.

#### 3.7.3. Representative Cases

Figure 6 illustrates representative cases on how structural parameters can correlate to functional parameters.

Five representative patients were selected to illustrate a range of findings in a structure vs. functional outcomes from more to less severe grades of VF defects (left to right) correlated to structural parameters (GC-IPL). The first two cases (A, B) show abnormal results in visual function (VA and VF) as well as pathological structural scans (GC-IPL thinning). In these cases, the patterns of VF defects corelate with GC-IPL thinning (bitemporal pattern) and localization of the tumor. More advanced changes are seen in the right eye in both cases. The third patient (case C) has normal visual acuity but less severe VF defects and structural abnormalities (thinning of GC-IPL) illustrating the same pattern of GC-IPL thinning (bitemporal pattern). The last two cases represent a patient with GC-IPL thinning (complete homonymous pattern) with mild reduction in VF parameters (case D) as well as a patient with GC-IPL thinning (binasal pattern) without any VF defects (case E). In these latter cases, structural changes are more obvious compared to the VF results. 

##### Case A

Nine-year-old previously healthy girl without criteria for NF1 being assessed for persisting, severe headache and impaired vision of her right eye. Neurological examination was normal. Ophthalmological findings included: VA RE 0.5 and LE 1.25; right positive relative afferent pupil defect (RAPD); Goldmann VF with left homonymous hemianopia and right optic disc pallor. Magnetic resonance imaging demonstrated hydrocephalus and a contrast-enhancing lesion in the chiasm-hypothalamus region with suprasellar growth extending to the cerebral peduncles. Treatment included partial resection and chemotherapy, initially according to the SIOP LGG 2004 protocol. Histopathology confirmed pilocytic astrocytoma grade I. When she entered the study at age 12, MRI demonstrated bilateral intracranial optic nerve, chiasmatic and right post-chiasmatic tumor involvement. Ophthalmological data including VF and OCT at study entry are presented in Figure 6. 

##### Case B

Eight-year-old, prematurely born and previously healthy boy without criteria for NF1, was assessed by a pediatric neurologist due to an episode of seizures in school. He had a history of headache for many years. Neurological examination was normal. Ophthalmological findings included: VA RE 0.1 and LE 1.0; right RAPD; Goldmann visual field with left homonymous hemianopia; right optic disc pallor. Magnetic resonance imaging demonstrated right optic nerve and right chiasm involvement. He underwent partial tumor resection. Histopathology confirmed pilocytic astrocytoma grade 1. When he entered the study at age 17, MRI demonstrated right-sided intracranial optic nerve and chiasmatic tumor involvement. Ophthalmological data including VF and OCT at study entry are presented in Figure 6. 

##### Case C

Eight-year-old boy with NF1 underwent MRI due to suspected pathological VF showing involvement of the left (intracanalicular and intracranial part) and right (intracranial) optic nerves, chiasm, left post-chiasmatic optic tract, and lateral geniculate nucleus. Neurological examination was normal. Ophthalmological findings included: VA RE 0.9 and. LE 0.8, Goldmann VF with bitemporal defects and nasal defect in the left eye as well as pallor of the left optic nerve. Chemotherapy was initiated, initially according to the SIOP LGG 2004 protocol. At study entrance at age 14, MRI findings were unchanged, except for additional involvement of the left lateral geniculate nucleus. Ophthalmological data including VF and OCT at study entry are presented in Figure 6. 

##### Case D

One-year-old boy, normally developed without criteria for NF1 was assessed due to absence seizures. He had a normal visual behavior according to age. There was no strabismus or signs of optic disk edema. Magnetic resonance imaging demonstrated tumor growth in the right temporal lobe. Treatment included subtotal extirpation. Histopathology showed pilocytic astrocytoma grade 1. When he entered the study at age 7, MRI demonstrated a tumor residue in the right temporal lobe. Ophthalmological data including VF and OCT at study entry are presented in Figure 6. 

##### Case E

Five-year-old girl with NF1 was assessed due to left sensorineural hearing loss and precocious puberty (accelerated growth). Ophthalmological findings were: VA RE 0.8 and LE 0.8 with normal optic nerves and Goldmann VF. Magnetic resonance imaging demonstrated tumor growth in both intracranial optic nerves as well aschiasm and hypothalamus. No treatment was indicated. Upon study entrance at age 9, MRI was unchanged. Ophthalmological data including VF and OCT at study entry are presented in Figure 6. 

## 4. Discussion

Based on the results from this study, we believe that OCT could add valuable insight when examining children at risk of developing vision-threatening changes due to OPG. The OCT examinations proved to be well tolerated and possible to perform in children as young as three to four years of age. In order to evaluate the usefulness of OCT, outcome measures were compared with standard examinations for this condition, i.e., VA, VF, and MRI. The results showed that GC-IPL thinning, as displayed with OCT, strongly correlated with VA, VF deficits, and MRI findings in this setting. 

Average GC-IPL thickness analysis accurately discriminated between eyes with or without OPG, as has been similarly reported by Gu et al. in children with OPG. A positive correlation between average GC-IPL and VA in patients with OPG (sporadic or NF1 associated) has also been obtained by Hepokur et al. [34]. Furthermore, average GC-IPL thinning was correlated with reduced VFI sensitivity (*p* < 0.001). In eyes with MRI-verified OPG that underwent all examinations, OCT showed abnormal GC-IPL results in 83% of cases, while abnormal VF patterns and VA was seen in 65% and 18% of cases, respectively. OCT identified all eyes with reduced vision based on VA or VF. It has been shown to be sensitive to early neuro-retinal degeneration in patients with pre-perimetric glaucoma [35], in children with diabetes type 1 without diabetic retinopathy [36] and in patients with chiasmal compression caused by brain tumors [37]. It is therefore possible that OCT has a higher sensitivity to detect retinal degeneration caused by OPG compared to functional tests.

Due to lack of a normality database for children in the OCT software, limits for abnormality were set according to a previous study of healthy children [29]. In addition, we developed a qualitative grading system to facilitate comparison of GC-IPL topography and VF to the localization of the tumor on MRI. 

The GC-IPL pattern loss corresponded well to VF defects (completely in 60% and partly in 35%) as well as MRI findings (92%). Similar correlations have been observed in patients with, for example, multiple sclerosis [38,39]. Moreover, an example of thinning of GC-IPL corresponding to VF in a child with OPG has been presented by Gu et al. [30]. Linking visual function to ocular structure is of paramount importance and since evaluation of tumor progression on MRI is often difficult, treatment relies heavily on clinical findings (VA and ophthalmoscopy). Being subjective in its nature, VA is not always easily obtained nor interpreted and normative values vary widely in young children. In this study, we found significant correlations between VA and OCT results, while OCT was shown to be more sensitive. Satisfactorily, children with normal GC-IPL also had normal VA. Therefore, we conclude that in cases of unreliable VA testing, normal GC-IPL result might confirm intact visual pathways. Nevertheless, GC-IPL analyses should always be interpreted by an ophthalmologist alongside a full ocular examination to minimize the risk of misinterpretation. For instance, normal congenital variations in ocular size or refraction such as myopia [29] can cause variations in GC-IPL thickness. In cases of children with normal VA and GC-IPL thinning without known myopia, an underlying pathology must be ruled out. Furthermore, a progressive decline in GC-IPL thickness if children with normal VA may emphasize the need for radiologic investigation. 

When the tumor affects the visual pathway, retrograde and sometimes transsynaptic degeneration occurs and can be objectively detected by OCT, possibly before functional loss can be detected using conventional perimetry and VA testing. In that manner, GC-IPL thinning could be a predictor of VF changes [38]. It is well established that loss of retinal ganglion cells can be demonstrated prior to detectable visual loss in chronic optic neuropathies. For example, in glaucoma, a common chronic ocular disorder leading to bilateral optic neuropathies, VF defects do not appear until at least 30% of retinal ganglion cells have died [39]. In the current study there were three eyes (5%) with thin GC-IPL and normal visual function possibly indicating pre-perimetric (early structural changes in the optic pathway without corresponding functional changes in standard automated perimetry) retinal ganglion cell loss. Retinal ganglion cell layer analysis can therefore be considered as a biomarker of early structural loss when VF and VA are normal. 

Visual field examination is valuable in neuroophthalmological conditions and provide information about lesion localization and extent. However, VF examinations demand a high degree of patient compliance in order to provide reliable results. For example, the patient has to look straight forward at least five minutes per eye, pay attention to what is happening in the periphery, and at the same time suppress the reflex to move the gaze. Young children usually have difficulties keeping fixation for longer periods of time. In such cases, poor cooperation can mimic pathological changes or mask existing defects. Optical coherence tomography also requires good fixation but during a much reduced time frame (about two seconds) and does not require psychophysical involvement in the same way as perimetry. Thus, a main advantage of this technology is fast and objective acquisition of data. 

By combining functional measurements with OCT findings, clinical examination becomes safer and more reliable. Visual field examination can be very demanding for children and many times reliable results are not achievable. If structural changes seen in the eye as measured with OCT can predict VF outcome, this could be a very useful method. By improving clinical follow-up of OPGs, treatment choices can be optimized and prevent further visual loss and, in the worst case, blindness. The examinations are somewhat more time-consuming than ordinary follow-ups, but at the same time patient-friendly and not painful. Moreover, OCT can be performed in a clinical setting and is a quick and inexpensive way to monitor and follow patients compared to MRI. 

Optical coherence tomography provides useful information serving as a link between visual function tests and MRI in children at risk of developing vision loss due to OPG. It offers objective measures and parameters that detect retinal degeneration caused by damage to the visual pathways. Finally, GC-IPL topography correlates with visual function and location of brain tumors. 

## 5. Conclusions

Our study indicates that GC-IPL measures could serve as an early marker of vision-threatening changes related to OPG and as a valuable link between MRI and visual function tests. Thinning of GC-IPL and differences in topography between eyes are strong indicators of and predictive of vision loss related to OPG.

## 6. Limitations

It is not completely understood how long it takes for brain damage engaging the visual pathways to show pathological thinning on OCT. The structure–function relationship between GC-IPL and visual function is somewhat imperfect and depends on disease stage. At a very early stage, it is possible that retinal structure is affected before functional loss is detectable using conventional perimetry and VA testing. At a moderate stage, correlation may be more obvious and at a late stage the visual loss progresses although OCT is unable to detect further structural loss (i.e., floor effect). In adults, Bowd et al. found a floor effect for GC-IPL between 31 and 45 µm [40]. Another limitation in this study is the relatively small comparison group which is due to the limited number of patients with unilateral OPG. 

## Figures and Tables

**Figure 1 cancers-14-00318-f001:**
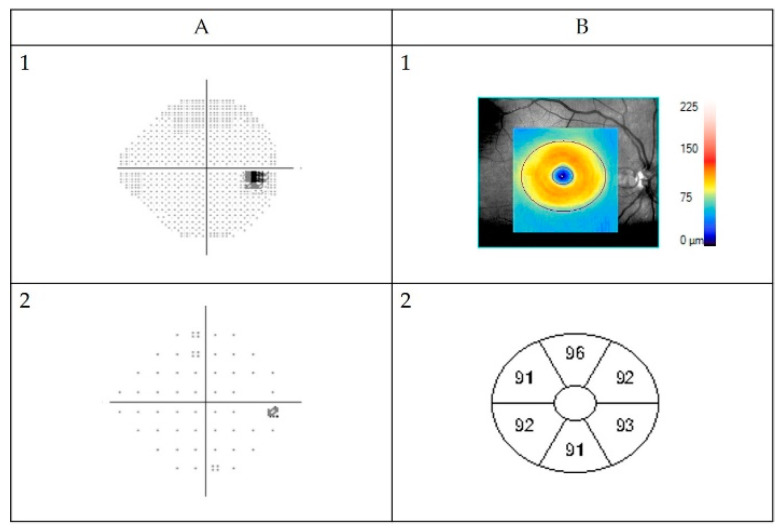
Example of normal automated Humphrey Field Analyzer visual field (VF) measurement (**A**) and GC-IPL (ganglion cell + inner plexiform layer) scan obtained by Zeiss Cirrus high-definition optical coherence tomography (**B**) from a healthy right eye. (**A1**) Visual field presented in grey scale, black indicating visual field defects. The black spot on the right side is the normal “blind spot”, representing the location of the optic nerve head, 15° nasally in the retina. Below pattern deviation plot (**A2**). This is in fact a plot of the probability of each point change being normal. If less than 0.5% (indicated by the solid black squares), then the point change is highly unlikely to be normal. Visual field sensitivity given for each eye as percentage of intact visual field (Visual Field Index). (**B1**) Color-coded topography of the macular GC-IPL layer with blue color indicating thinning. Below (**B2**) sectoral maps showing macular GC-IPL thicknesses in superiotemporal, superior, superionasal, inferionasal, inferior, and inferiotemporal sectors. Average GC-IPL values as well as minimum GC-IPL for each eye (microns).

**Figure 2 cancers-14-00318-f002:**
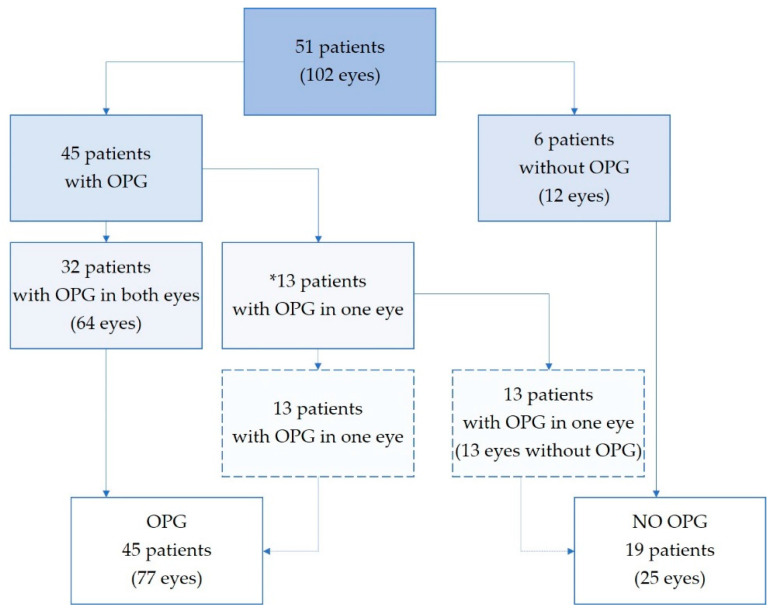
Study population. Eyes stratified in subgroups depending on optic pathway glioma involvement. *13 patients had strictly one-sided optic nerve glioma, leaving one eye unaffected. OPG—optic pathway glioma.

**Figure 3 cancers-14-00318-f003:**
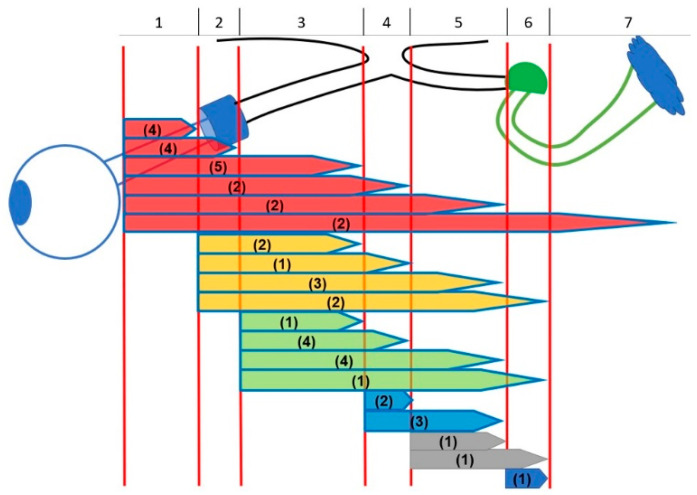
Different anatomic locations of optic pathway gliomas. Extension of tumors: (1) intraconal (posterior boundary orbital apex); (2) intracanalicular (posterior boundary optic foramen); (3) intracranial-prechiasmatic; (4) chiasmatic; (5) optic tract; (6) lateral geniculate nucleus; (7) optic radiation. Number of patients in brackets (*n* = 45).

**Figure 4 cancers-14-00318-f004:**
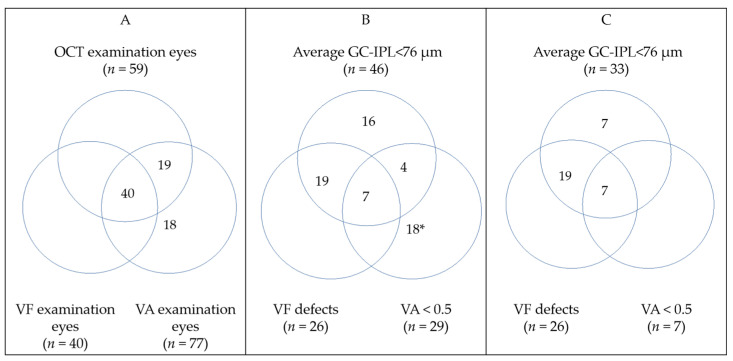
Venn diagrams illustrating: (**A**) All eyes with optic pathway glioma (OPG) that underwent any ophthalmological examination; (**B**) all eyes with OPG that had any abnormal ophthalmological finding; (**C**) eyes with OPG that underwent all ophthalmological examinations and had pathological findings. Detailed information about clinical finding may be found in Appendix A. * None of those eyes provided OCT data (14 blind). GC-IPL—ganglion cell + inner plexiform layer; VA—visual acuity; VF—visual field.

**Figure 5 cancers-14-00318-f005:**
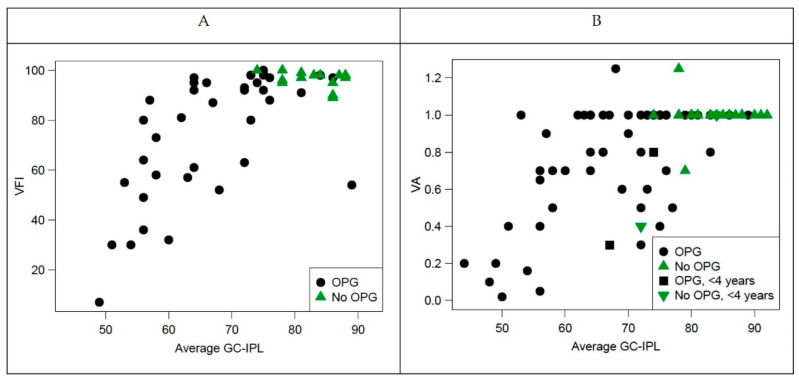
Correlation between visual function (visual field index and visual acuity and structure (average ganglion cell-internal plexiform layer [GC-IPL]). (**A**) Visual field index plotted against average GC-IPL; (**B**) visual acuity plotted against average GC-IPL. OPG—optic pathway glioma; VFI—visual field index.

**Figure 6 cancers-14-00318-f006:**
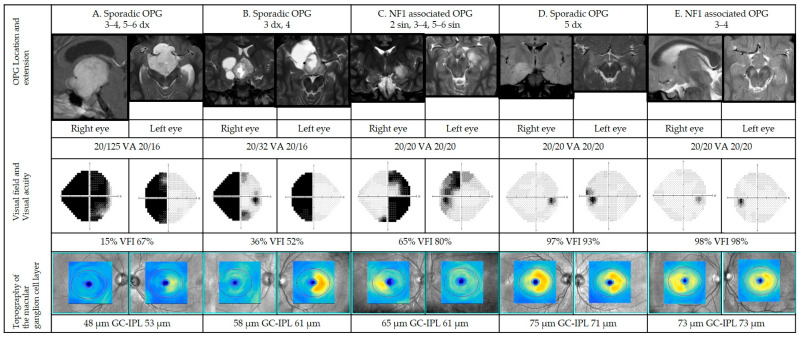
Representative cases of patients with optic pathway gliomas (OPGs) in different anatomic locations (please see Figure 1; dx: right side, sin: left side). Magnetic resonance imaging (MRI), visual acuities, visual fields, and optical coherence tomographies for cases **A**–**E**. For each case, OPG is presented on representative MRI. Visual fields are presented in gray scale, black indicating visual field defects. Sensitivity as percentage of the intact visual field (visual field index). Below, color-coded topography of the macular ganglion cell-internal plexiform layer (GC-IPL), blue color indicating thinning. Average GC-IPL values for each eye in microns. GC-IPL—ganglion cell-internal plexiform layer; VA—visual acuity; VFI—visual field index. Dx—right; sin—left.

## Data Availability

The datasets analyzed during the current study are available from the corresponding author on reasonable request.

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
