# Peer review of "Optical Coherence Tomography Identifies Visual Pathway Involvement Earlier than Visual Function Tests in Children with MRI-Verified Optic Pathway Gliomas"

_cancers, 2022, doi:10.3390/cancers14020318_

Round 1

Reviewer 1 Report

Very interesting paper, very well carried out. Congratulations.Just 3 comments:
1) I think it would be interesting to add a table recompiling and specifying the data of the 33 cases referred to in figure 4C. (Anatomic location, VF affected, GCL thickness not only the mean but also nasal and temporal sectors…). Maybe as supplementary material.
2) page 4 line 165-168: Was the pathological GCL definition based on any previous work (different from the normal database)?3) page 13 lines 476-477: Could you provide any number from which GCL would not add any significant information because of the floor effect?

Author Response

Comments and Suggestions for Authors

Very interesting paper, very well carried out. Congratulations.Just 3 comments:
1) I think it would be interesting to add a table recompiling and specifying the data of the 33 cases referred to in figure 4C. (Anatomic location, VF affected, GCL thickness not only the mean but also nasal and temporal sectors…). Maybe as supplementary material.

Response: Thank you for the suggestion! We have added a Table S1 as supplementary material and now refer to this in the Figure 4C legend (page 15-15, line 508-513)

Supplementary material:

Table 1. Clinical findings in 33 eyes with OPG that underwent all ophthalmological examinations and had pathological findings.

No Patient, Eye (R/L)

VA

Average GC-IPL thickness (µm)

Combined nasal sectors  thickness (µm)

Combined temporal sectors thickness (µm)

VF affected

OPG location

1R

0,65

56

56,5

53

Nasal complete and temporal partial

3-4

1L

1,25

68

52

87

Temporal complete

4

2R

0,05

56

54

57,5

Temporal partial and superior nasal partial

3-5

2L

0,7

56

55,5

58

Temporal partial and nasal inferior partial

3-4

3R

0,6

73

65

82

Inferior temporal quadrant

3-4

3L

0,8

74

65

82

Normal

3-4

4R

1

66

59,5

72,5

Superior temporal partial

3-5

4L

1

64

60

65,5

Normal

3-5

5R

1

75

76,5

78

Normal

4

5L

1

75

72

80,5

Normal

4

6R

1

64

56,5

75

Inferior temporal complete and superior temporal partial

3-4

6L

1

62

60

63,5

Temporal partial and superior nasal partial

2-6

7R

0,4

56

53,5

61,5

Complete nasal and superior temporal partial

5-6

7L

0,7

60

61,5

54,5

Complete temporal and superior nasal partial

5-6

8R

0,2

49

48,5

47,5

Complete nasal and subtotal temporal

3-6

8L

1

53

42,5

65

Temporal complete

3-4

9R

0,3

72

66,5

77,5

Partial temporal and nasal

3-4

10R

0,7

58

58

55,5

Complete superior nasal, partial inferior nasal and partial superior temporal

1

11R

1

75

74,5

77

Normal

3-4

11L

1

75

74,5

77,5

Normal

4

12R

0,7

64

61

69

Enlarged blind spot

2-5

12L

0,8

64

57

73

Normal

2-5

13R

0,4

75

71,5

80,5

Partial temporal

4

13L

0,8

72

69,5

76,5

Partial temporal 

1-5

14L

1

63

45

81

Complete temporal

4-5

15L

0,5

58

59,5

57

Diffuse central

1-2

16R

0,4

51

51

52,5

Central scotoma and complete temporal defect

3-4

17R

1

73

68,5

79,5

Normal

3-4

17L

1

73

68,5

79,5

Normal

3-4

18L

1

72

63

81,5

Normal

5

19R

1

67

64,5

73

Partial temporal 

1-7

20R

0,9

57

54

62

Partial temporal

4

20L

0,16

54

55

52,5

Complete inferior temporal, partial superior temporal and inferior nasal temporal

1-4

Extension of tumors: (1) Intraconal (posterior boundary orbital apex); (2) Intracanalicular (posterior boundary optic foramen); (3) Intracranial-prechiasmatic; (4) Chiasmatic; (5) Optic tract; (6) Lateral geniculate nucleus; (7) Optic radiation. VA=visual acuity, L=left eye, R=right eye.

2) page 4 line 165-168: Was the pathological GCL definition based on any previous work (different from the normal database)?

Response: Thank you for your comment! The pathological GCL definition was based on our previous study where we presented a normative database among healthy 6.5 years old Swedish children (Arnljots et al.). We have presented it on page 5 line 178-180 in the material and methods section. To the best of our knowledge, there are no published detailed criteria concerning pathological GC-IPL thickness in children.

Arnljots, U., et al., Profile of macular ganglion cell-inner plexiform layer thickness in healthy 6.5 year- old Swedish children. BMC Ophthalmol, 2020. 20(1): p. 329.3) page 13 lines 476-477: Could you provide any number from which GCL would not add any significant information because of the floor effect?

Response: Thank you for your valuable comment. We have now added this information on page 13 line 503- 504 as well as a reference.  In adults, Bowd et al.  found a floor effect for GC-IPL between 31-45µm.

Bowd C, Zangwill LM, Weinreb RN, Medeiros FA, Belghith A. Estimating Optical Coherence Tomography Structural Measurement Floors to Improve Detection of Progression in Advanced Glaucoma. Am J Ophthalmol. 2017 Mar;175:37-44. doi: 10.1016/j.ajo.2016.11.010. Epub 2016 Nov 30. PMID: 27914978; PMCID: PMC5337134.

Reviewer 2 Report

I think that this study adds to the gathering data about supplementing current modalities of VA, VF, OCT of the RNFL, and MRI with information from OCT of the GC-IPL, especially earlier changes and greater ease in very young children. The control-comparison group without OPG is smaller, perhaps out of necessity, but maybe that should be acknowledged. The points made are clear though somewhat wordy and sometimes repetitive and could possibly be tightened up. But I feel that these were important findings in the particular setting of OPG where monitoring plays such a key role in management. I have no specific recommendations to make.

Author Response

I think that this study adds to the gathering data about supplementing current modalities of VA, VF, OCT of the RNFL, and MRI with information from OCT of the GC-IPL, especially earlier changes and greater ease in very young children. The control-comparison group without OPG is smaller, perhaps out of necessity, but maybe that should be acknowledged. The points made are clear though somewhat wordy and sometimes repetitive and could possibly be tightened up. But I feel that these were important findings in the particular setting of OPG where monitoring plays such a key role in management. I have no specific recommendations to make.

Response: Thank you for your suggestions. We have now added one statement on the small comparison group in the limitation section, page 13, line 504-506. We have also shortened the manuscript by decreasing the repetitions.

Reviewer 3 Report

Manuscript Title: Optical coherence tomography identifies visual pathway involvement earlier than visual functional tests in children with 3 MRI-verified optic pathway gliomas

This is an interesting paper describing how structural OCT can be used as a targeted tool to objectively measure clinical progression of ganglion cell loss in children optic pathway gliomas (OPGs). The Authors reported how thinning of GC-IPL is strong indicator of predictive vision loss related to OPG.

  • Grammatical English is appropriate

Methods

2.4. GC-IPL layer

  • Please add reference to the sentence: “Since pRNFL has been shown to be less accurate than GC-IPL we only used the latter in the analyses.”

Discussion:

  • Nothing to add

Figures and tables

  • Figures and tables are suitable

Author Response

Manuscript Title: Optical coherence tomography identifies visual pathway involvement earlier than visual functional tests in children with 3 MRI-verified optic pathway gliomas

This is an interesting paper describing how structural OCT can be used as a targeted tool to objectively measure clinical progression of ganglion cell loss in children optic pathway gliomas (OPGs). The Authors reported how thinning of GC-IPL is strong indicator of predictive vision loss related to OPG.

  • Grammatical English is appropriate

Methods

2.4. GC-IPL layer

  • Please add reference to the sentence: “Since pRNFL has been shown to be less accurate than GC-IPL we only used the latter in the analyses.”

Response: Thank you for your comment. The following reference is now added on page 5, line 178-180:

It is proposed that GC-IPL is a more accurate and reliable biomarker of vision, because it is not confounded by axonal swelling, axonal atrophy, and blood vessel artifacts as in pRNFL analysis (Gu S et al.).

Gu S, Glaug N, Cnaan A, Packer RJ, Avery RA. Ganglion cell layer-inner plexiform layer thickness and vision loss in young children with optic pathway gliomas. Invest Ophthalmol Vis Sci. 2014;55(3):1402-8).

Discussion:

  • Nothing to add

Figures and tables 

  • Figures and tables are suitable

Reviewer 4 Report

The authors investigates whether optical coherence tomography (OCT) could add useful 31 information in examination of children with optic pathway glioma (OPG) and at high risk of developing vision loss. For this purpose, the relationship between ganglion cell- inner plexiform layer 33 (GC-IPL) thickness and visual function, evaluated with tests of visual acuity (VA) and visual field 34 (VF), as well as tumor site according to MRI, were examined in 35 children with OPG.

The results clarelly indicates that GC-IPL measures could serve as an early marker of vision threatening changes related to OPG and as a valuable link between MRI and visual function tests.

The paper is well written and the research design appropriate.

I have only some comments.

1) the authors state that "The diagnosis of OPG was determined using contrast-enhanced MRI and, if possible, 108 with biopsy and histopathological analysis." biopsy is not the standard method for the diagnosis of OPG. The possibility of make a biopsy did not indicate that having a biopsy is suggested. Please change this statement.

2) The mean age at study entrance was 9.3 years. At this age the collaboration of a young child is good, but most commonly the diagnosis of OPG is at 3-4-5 years of age. In younger children the use of a macular map by OCT may be difficult. Some authors suggest using RNFL thickness by OCT, that is more rapid (Acta Ophthalmol. 2018 Dec;96(8):e1004-e1009;  Invest Ophthalmol Vis Sci. 2013 Dec 17;54(13):8112-8.). Please discuss this option in the discussion section.

Author Response

The authors investigates whether optical coherence tomography (OCT) could add useful 31 information in examination of children with optic pathway glioma (OPG) and at high risk of developing vision loss. For this purpose, the relationship between ganglion cell- inner plexiform layer 33 (GC-IPL) thickness and visual function, evaluated with tests of visual acuity (VA) and visual field 34 (VF), as well as tumor site according to MRI, were examined in 35 children with OPG.

The results clarelly indicates that GC-IPL measures could serve as an early marker of vision threatening changes related to OPG and as a valuable link between MRI and visual function tests.

The paper is well written and the research design appropriate.

I have only some comments.

1) the authors state that "The diagnosis of OPG was determined using contrast-enhanced MRI and, if possible, 108 with biopsy and histopathological analysis." biopsy is not the standard method for the diagnosis of OPG. The possibility of make a biopsy did not indicate that having a biopsy is suggested.

Please change this statement.

Response: Changed to (page 3 line 108-110):

In cases where surgery was performed biopsy and histopathological analysis were undertaken.

2) The mean age at study entrance was 9.3 years. At this age the collaboration of a young child is good, but most commonly the diagnosis of OPG is at 3-4-5 years of age. In younger children the use of a macular map by OCT may be difficult. Some authors suggest using RNFL thickness by OCT, that is more rapid (Acta Ophthalmol. 2018 Dec;96(8):e1004-e1009;  Invest Ophthalmol Vis Sci. 2013 Dec 17;54(13):8112-8.). Please discuss this option in the discussion section.

In this study both GC-IPL and pRNFL thicknesses (not requiring application of dilatation drops) were examined, however detailed information was only given regarding GC-IPL results.  

“Since pRNFL has been shown to be less accurate than GC-IPL we only used the latter in the analyses. It is proposed that GC-IPL is a more accurate and reliable biomarker of vision, because it is not confounded by axonal swelling, axonal atrophy, and blood vessel artifacts as in pRNFL analysis [33]”. Page 5, line 178-180.

Response: In the studies by Parrozzani et al. only pRNFL (after pupil dilatation) was analyzed.  This fact makes it harder to compare both methods.  In our opinion, GC-IPL results were generally easier to obtain at an earlier age than the pRNFL measurements. Also, there were fewer motion-artifacts /segmentation errors among GC-IPL scans. One explanation could be that the children found it easier to fixate straight forward when measuring GC-IPL compared to slightly sideways when measuring pRNFL. Our intention is to present the results of our comparison between pRNFL and GC-IPL findings in the next paper.
